# Effect of an Electronic Alert System on Hepatitis B Virus Reactivation in Patients Receiving Immunosuppressive Drug Therapy

**DOI:** 10.3390/jcm11092446

**Published:** 2022-04-26

**Authors:** Akira Asai, Saho Hirai, Keisuke Yokohama, Tomohiro Nishikawa, Hiroki Nishikawa, Kazuhide Higuchi

**Affiliations:** 1The Second Department of Internal Medicine, Osaka Medical and Pharmaceutical University, Takatsuki 569-8686, Japan; keisuke.yokohama@ompu.ac.jp (K.Y.); tomohironisikawa5795@yahoo.co.jp (T.N.); hiroki.nishikawa@ompu.ac.jp (H.N.); kazuhide.higuchi@ompu.ac.jp (K.H.); 2Faculty of Medicine, Osaka Medical and Pharmaceutical University, Takatsuki 569-8686, Japan; rmtw4947@yahoo.co.jp

**Keywords:** HBV reactivation, electronic alert, immunosuppressive drug therapy

## Abstract

Hepatitis B virus (HBV) reactivation (HBVr) can occur in patients receiving immunosuppressive drug therapies, causing significant morbidity and mortality. Although the guidelines for HBVr have been proposed by several academic societies, some providers do not follow them, resulting in HBVr and death. As HBV-DNA levels increase before liver enzyme levels do, we previously constructed an electronic alert system that recommends the measurement of HBV-DNA. Here, we investigated whether this alert system improves the HBV-DNA measurement rate and elicits responses according to guidelines. A total of 5329 patients were divided into two groups, before and after the introduction of the alert system, and the HBV-DNA measurement rates in both groups were compared. Because of the introduction of the alert system, the HBV-DNA measurement rate among HBsAg-negative patients with anti-HBs and/or anti-HBc before immunosuppressive drug therapy improved significantly. The HBV-DNA monitoring rate within 3 months also improved significantly (*p* = 0.0034) in HBV-remission phase patients. HBVr was detected immediately, and the affected patients were treated with nucleotide analogs before severe hepatitis onset. The introduction of the alert system for HBVr improved the HBV-DNA measurement rates in patients receiving immunosuppressive drug therapy, leading to the rapid treatment of patients with HBVr.

## 1. Introduction

Hepatitis B virus (HBV) belongs to the Hepadnaviridae family of small, enveloped, hepatotropic, partially double-stranded DNA viruses [1]. HBV is transmitted perinatally or horizontally via blood or genital fluids [2]. According to the World Health Organization, an estimated 296 million people were living with chronic HBV infection in 2019, with 1.5 million new infections reported each year [3]. Among the long-term complications of HBV infections, advanced liver diseases, such as cirrhosis and hepatocellular carcinoma, develop in a subset of individuals, causing high morbidity and mortality [4]. HBV results in an estimated 820,000 deaths due to these complications each year [5].

Acute HBV infection is usually subclinical in adults. Immunocompetent adults with acute HBV infection change spontaneously, with hepatitis B surface antigen (HBsAg) seroconversion to an antibody against hepatitis B surface antigen (anti-HBs), and only 5% of these adults develop chronic HBV infection. However, the risk of developing chronic HBV infection is 90% if the infection occurs at birth and 16–30% if it occurs in childhood [6,7]. In Japan, nucleoside and nucleotide analog oral reverse transcriptase inhibitors (NUCs) have been approved for the treatment of chronic hepatitis B since 2000, and they have revolutionized the management of chronic hepatitis B [8,9]. However, HBsAg clearance is very rare (0–5%) even after prolonged treatment, and frequent viral rebound upon therapy withdrawal indicates the need for lifelong treatment [10].

HBV reactivation (HBVr) is characterized by the reappearance of HBV in patients who were previously resolved or an increase in HBV viremia in patients with inactive chronic hepatitis [11]. According to acute infection reports on HBVr with HB seroconversion in patients receiving immunosuppressive therapy, HBV persists in the liver up to more than 10 years after acute HBV infection [12]. This reactivation can occur spontaneously in HBV patients, but it is more commonly triggered by immunosuppressive drug therapies against underlying diseases, such as cancers or collagen diseases [13,14,15,16]. Hepatitis due to HBVr in these patients is easily exacerbated and can cause significant morbidity and mortality [11,17]. In addition, the prognosis of these patients deteriorates due to the interruption or discontinuation of treatment for underlying diseases [18,19]. The virological key to this reactivation is an intracellular HBV replication intermediate, called covalently closed circular DNA, which resides in the nucleus of infected cells as an episomal plasmid-like molecule that gives rise to progeny viruses [20,21].

Guidelines on the prevention and treatment of HBVr during immunosuppressive drug therapy have been published [22,23,24]. If serum HBsAg is positive in patients before starting immunosuppressive drug therapy, NUC treatment against HBV should be initiated. Even if serum HBsAg is negative, HBV-DNA needs to be measured in patients with serum anti-HBs or antibodies to hepatitis B core antigen (anti-HBc). When HBV-DNA is detected in these patients, treatment with NUCs should be initiated. Moreover, even if HBV-DNA is not detected in these patients, its monitoring is needed every 1–3 months. Despite these guidelines, deaths caused by HBVr are reported [25,26,27]. Clinical parameters of HBV may not have been noticed in these patients before and during immunosuppressive drug therapy. Some reports on electronic alerts for the screening of hepatitis C virus (HCV) and HBV infection are available [28,29]; however, reports on electronic alerts for HBVr are insufficient [30,31].

The measurement of HBV-DNA is one of the most important factors in the prevention and treatment of HBVr. Some studies have recommended the measurement of HBV-DNA levels when monitoring HBVr [32,33,34]. The reason is that HBV-DNA levels increase before the development of hepatitis due to HBVr [17]. Therefore, we constructed an electronic alert system to measure HBV-DNA against HBVr. HBV-DNA was measured in three types of patients receiving immunosuppressive drug therapy: (1) HBsAg-positive patients, (2) HBsAg-negative patients with anti-HBs and/or anti-HBc before immunosuppressive drug therapy, and (3) HBV remission patients (patients negative for HBV-DNA in HBsAg-negative patients with anti-HBs and/or anti-HBc) with HBV-DNA monitoring during immunosuppressive drug therapy (Figure 1). We investigated whether this system improves the HBV-DNA measurement rate among patients and elicits responses according to the guidelines.

## 2. Patients and Methods

### 2.1. Patients

All procedures performed in this study were in accordance with the ethical standards of the institution, ethical guidelines for medical and human subjects in Japan, and the 1964 Helsinki Declaration and its later amendments. This retrospective study was approved by the Ethical and Scientific Committee of the Osaka Medical and Pharmaceutical University (IRB approval number: 2020–074). A total of 5329 patients who received immunosuppressive drug therapies between August 2015 and April 2020 were enrolled in this study at the Osaka Medical and Pharmaceutical University Hospital (Table 1). This included 3462 patients with advanced-stage cancer who underwent chemotherapy and 1867 patients who underwent other immunosuppressive therapies. Of these, 991 were treated in urology, 890 patients were treated in collagen diseases, 656 in otolaryngology, 654 in gastroenterology, 552 in obstetrics and gynecology, 335 in breast surgery, 245 in respiratory medicine, 233 in hematology, 213 in pediatrics, 188 in digestive surgery, 133 in dermatology, 105 in ophthalmology, and 114 in others. The patients were divided into two groups (before and after the introduction of the alert system), and the background factors of these patients were compared (Table 1).

### 2.2. Reagents

In patients who received immunosuppressive drug therapy at the Osaka Medical and Pharmaceutical University Hospital, HBsAg, anti-HBs, and anti-HBc levels were measured before therapy using ARCHITECT^®^ HBsAg, Anti-HBs, and Anti-HBcII measurement kits (Abbott Japan Co., Ltd., Chiba, Japan), respectively, that employ the chemiluminescence immunoassay (CLIA) method and these indicate a positive status at >0.05 IU/mL, >10.0 mIU/mL, and >1.0 S/CO, respectively. COBAS^®^ TaqMan^®^ HBV “auto” Version 2.0 kit that employs the real-time PCR method (Roche Diagnostics, Tokyo, Japan) was used for the measurement of HBV-DNA, and values above the detection sensitivity were considered positive. In this study, HBVr was defined as an increase in HBV-DNA levels above 1.3 logIU/mL. Clinical background data of patients with HBVr were retrospectively extracted from the medical records.

### 2.3. The Electronic Alert System for HBVr

Based on the results of the HBsAg, anti-HBs, and anti-HBc tests, patients requiring the measurement of HBV-DNA were classified into three groups. The first group was patients who were positive for the HBsAg test (HBsAg-positive patients). The second group was patients who were negative for the HBsAg test and positive for either or both of the anti-HBs test and anti-HBc test (HBsAg-negative patients with anti-HBs and/or anti-HBc). When HBV-DNA was measured in HBsAg-negative patients with anti-HBs and/or anti-HBc, these patients with negative HBV-DNA were defined as the remission phase of HBV-DNA (HBV-remission phase patients). According to the guidelines for the prevention and treatment of HBVr during immunosuppressive drug therapy provided by the Japan Society of Hepatology [35], HBV-DNA monitoring should be performed every 1–3 months for patients undergoing immunosuppressive drug therapy. The third group included patients in the HBV-remission phase who were not monitored for HBV-DNA within 3 months after the introduction of immunosuppressive drug treatment. We constructed a system of electronic alerts to recommend the measurement of HBV-DNA. For patients in each group, alerts were automatically displayed on the front page of the patient’s electronic medical record. This system was put into operation on electronic medical records at our hospital in August 2019. We investigated the improvement caused by introducing this alert system in HBV-DNA measurement rates of the three groups of patients.

### 2.4. Statistical Analysis

Clinical laboratory values were not normally distributed; therefore, the Mann–Whitney *U* test was used to analyze continuous scales. Fisher’s exact test was used to analyze nominal scales. All the recorded *p*-values were two-sided, and differences were considered significant at *p* < 0.05. All analyses were performed using the JMP software version 13 (SAS Institute Inc., Cary, NC, USA).

## 3. Results

### 3.1. Clinical Background before and after the Introduction of the Alert System

Patient characteristics at the time of initial HBsAg determination in all 5329 patients are shown in Table 1. The number of patients before the introduction of the system was 4417 and was 912 after the introduction of the system. Chemotherapy was administered to 2874 (65.1%) of 4417 patients before and to 588 (64.5%) of 912 patients after the introduction of the system, with no significant difference between the two groups. Anti-HBs and anti-HBc levels were immediately measured in patients who were negative for HBsAg. A total of 917 (20.8%) out of 4417 patients tested positive for either or both antibodies before and 219 (21.3%) of 912 patients after the introduction of the system, with significantly more patients in the post-system group (*p* = 0.0330). Patients in the HBV-remission phase (HBV-DNA negative in HBsAg-negative patients with anti-HBs and/or anti-HBc) were 704 of 4417 (15.9%) prior to and 188 of 912 (20.6%) after the introduction of the system (*p* = 0.0008).

### 3.2. Effect of the Alert System on HBV-DNA Measurement Rates

Among HBsAg-positive patients, HBV-DNA was measured in 35 of 42 patients (83.3%) before and in 8 of 9 patients (88.9%) after the introduction of the system, with no significant difference between the two groups (Table 2). Based on these findings, HBV-DNA was measured in patients with positive HBsAg before and after the introduction of the alert system in our hospital.

Of the 5329 cases, those who were positive for HBsAg and HBV-DNA (*n* = 25) had a mean HBsAg of 2.30 ± 1.22 log_10_ IU/mL. Patients who were positive for HBsAg and negative for HBV-DNA (*n* = 18) had a mean HBsAg of 0.94 ± 1.73 log_10_ IU/mL, with significantly higher HBsAg levels in patients who were positive for HBsAg and HBV-DNA (*p* = 0.0045) (Figure 2).

Among those who were HBsAg-negative and positive for either one or both anti-HBs and anti-HBc 705 of 917 (76.9%) had their HBV-DNA measured prior and 188 of 219 (85.8%) after the introduction of the system, indicating significant improvement in the HBV-DNA measurements (*p* < 0.01). Only one patient in the group tested positive for HBV-DNA before the introduction of this system. This patient was immediately treated with entecavir and chemotherapy was safely performed. There were no HBV-DNA-positive individuals among HBsAg-negative patients with anti-HBs and/or anti-HBc in the group after the introduction of the system. Therefore, the introduction of the alert system improved the HBV-DNA measurement rate for HBsAg-negative patients with anti-HBs and/or anti-HBc. In addition, this alert system led to the treatment of HBV infection in a new patient.

The number of patients who underwent HBV-DNA monitoring within 3 months after immunosuppressive drug therapy was 305 out of 704 (43.3%) before and 104 out of 188 (55.3%) after the introduction of the system. The HBV-DNA measurement rate improved significantly after the introduction of the system (*p* < 0.01). Among them, 2 of 304 (0.7%) patients had HBV-DNA reactivation before and 0 (0%) after the introduction of the system. Therefore, the introduction of the alert system improved HBV-DNA monitoring rates in HBV-remission phase patients during the first 3 months after immunosuppressive drug therapy.

### 3.3. Two Cases of HBV-Reactivation in Patients with HBV-Remission Phase

The details of HBV-remission phase patients who experienced reactivation 3 months after immunosuppressive drug therapy detected by performing HBV-DNA monitoring are shown in Table 3. Two patients were not using rituximab. The first case was that of a 76-year-old man who underwent one course of gemcitabine and cisplatin therapy for renal cell carcinoma, and the HBV-DNA measured 84 days after the start of chemotherapy turned positive at 1.72 log_10_ IU/mL. Alanine aminotransferase (ALT) was slightly elevated (41 IU/L), and treatment with tenofovir alafenamide was initiated promptly. Two weeks later, HBV-DNA turned negative, liver damage improved, and chemotherapy was administered as planned. The second case was that of an 81-year-old man who underwent one course of gemcitabine and cisplatin therapy for bladder cancer, and the HBV-DNA measured on the 18th day after the start of chemotherapy turned positive to 1.75 log_10_ IU/mL. ALT was within the normal range, and entecavir treatment was started in this patient. Two weeks later, the HBV-DNA test was negative, and chemotherapy could be carried out as planned. In both cases, no significant liver damage was observed at the time of HBVr, and HBV-DNA also decreased rapidly after treatment with NUCs.

### 3.4. HBV-DNA Measurement Rates before and after the Introduction of the System in Patients Undergoing Chemotherapy

Hepatitis viral markers that were measured in 3462 patients prior to chemotherapy were present in 2874 patients before and in 588 patients after the introduction of the system (Table 4). The median age of patients with a 95% confidence interval was 63 (63.2–64.1) years before and 65 (64.1–66.0) years after the introduction of the system. The number of HBsAg-positive patients was 32 of 2874 (1.1%) before and 6 of 588 (1.0%) after the introduction of the system, with no significant difference between the two groups. The number of HBsAg-negative patients with anti-HBs and/or anti-HBc was 639 of 2842 (22.2%) before and 143 of 582 (24.3%) after the introduction of the system. No significant difference was observed between the groups. The number of patients in the HBV remission phase was 489 of 2874 (17.0%) before and 127 of 588 (21.6%) after the introduction of the system, with a significant increase (*p* = 0.0081). Among HBsAg-positive patients, the HBV-DNA was measured in 28 of 32 patients (87.5%) before and in 5 of 6 patients (83.3%) after the introduction of the system, with no significant difference between the two groups. HBV-DNA measurement rates were high in both groups. The number of HBV-DNA-positive patients among HBsAg-positive patients was 16 out of 28 (57.1%) before and 1 out of 5 (20.0%) after the introduction of the system. In HBsAg-negative patients with anti-HBs and/or anti-HBc, HBV-DNA was measured in 490 of 639 (76.7%) patients before and in 127 of 143 (88.8%) after the introduction of the system. The measurement rate improved significantly after using the system (*p* < 0.01). Among them, one patient tested positive for HBV-DNA before the introduction of the system. There were 179 of 489 (36.6%) patients in the HBV-remission phase who were monitored for HBV-DNA in the first 3 months after chemotherapy before and 65 of 127 (51.2%) after the introduction of the system, indicating a significantly improved monitoring rate for HBV-DNA (*p* < 0.01). In the group before the system, HBV-DNA reactivation was observed in two patients who were monitored for HBV-DNA. These findings indicate that the introduction of this system improved HBV-DNA measurement and monitoring in patients receiving chemotherapy.

### 3.5. HBV-DNA Measurement Rates before and after the Introduction of the System in Patients Undergoing other Immunosuppressive Therapies

Of the 1867 patients for whom HBV markers were measured before administering other immunosuppressive therapies, 1543 patients were from the group before, and 324 were from the group after the system was introduced (Table 5). No significant difference in sex or age between the two groups was observed. The number of HBsAg-positive patients was 10 of 1543 (0.7%) in the group before and 3 of 324 (0.9%) in the group after the introduction of the system; there was no significant difference between the two groups. The number of HBsAg-negative patients with anti-HBs and/or anti-HBc was 278 of 1543 (18.0%) in the group before and 76 of 324 (23.5%) in the group after the introduction of the system. The number of HBsAg-negative patients was significantly higher in the group after the introduction of the system (*p* = 0.02). The number of patients in the HBV remission phase was 215 of 1543 (13.9%) before and 61 of 324 (18.8%) after the introduction of the system, with no significant difference between the two groups. In HBsAg-positive patients, HBV-DNA was measured in 7 of 10 patients (70.0%) before and in 3 of 3 patients (100%) after the introduction of the system, with no significant difference among the groups. Among those with HBV-DNA measurements, 5 of 7 (71.4%) were positive for HBV-DNA in the group before the introduction of the system and 3 of 3 (100%) were positive in the group after the introduction of the system. In HBsAg-negative patients with anti-HBs and/or anti-HBc, HBV-DNA was measured in 215 of 278 patients (77.3%) before and in 61 of 76 patients (80.3%) after the introduction of the system, with no significant difference between the two groups, although both had a higher measurement rate. None of the patients were HBV-DNA-positive before or after the introduction of the system. HBV remission-phase patients who were monitored for HBV-DNA in the first 3 months after the introduction of other immunosuppressive therapies did not differ significantly in 126 of 215 (58.6%) patients before and in 39 of 61 (63.9%) patients after the introduction of the system. Among the HBV-remission patients who underwent HBV monitoring, no HBV-DNA-positive patients were found before or after the introduction of the system. Based on these results, the HBV-DNA measurement rate was high in patients undergoing other immunosuppressive therapies both before and after the introduction of the system.

## 4. Discussion

Previous reports show that in patients with HBV remission undergoing chemotherapy or immunosuppressive therapy, the HBVr rate varied from 1.9% to 41.5% depending on the diseases, reagents, and dose of glucocorticoids [36,37,38]. In our initial study, only two patients (0.7%) showed reactivation within the first 3 months of immunosuppressive drug therapy. This initial study did not include patients who presented reactivation after 3 months from the initiation of immunosuppressive drug therapy. Therefore, additional studies were conducted on patients who presented reactivation from 3 months to 6 years after introducing treatment for the underlying disease. A total of 409 patients could be repeatedly monitored for HBV-DNA every 3 months among patients who were able to continue immunosuppressive drug therapy for more than 3 months. We additionally investigated patients with HBVr after 3 months. As a result, two more patients were found to have HBVr after 3 months. The first patient was an 85-year-old man who underwent two courses of rituximab, cyclophosphamide, doxorubicin, vincristine, and prednisone therapy; four courses of rituximab and bendamustine therapy; and two courses of etoposide, prednisone, and rituximab therapy for diffuse large B-cell lymphoma and HBVr. HBV-DNA was converted to 1.95 log_10_ IU/mL at 596 days after the initiation of treatment. The patient’s serum ALT level was 13 IU/l, and entecavir treatment was promptly initiated. Two weeks later, HBV-DNA was negative, no hepatic disorder was observed, and chemotherapy was administered. The second patient, an 85-year-old woman, underwent six courses of docetaxel and cyclophosphamide therapy for ovarian carcinoma, which resulted in the conversion of HBV-DNA to 1.59 log_10_ IU/mL, measured 178 days after chemotherapy. The ALT level was 13 IU/l, and hepatic disorder was not recognized; entecavir treatment was promptly initiated. Two weeks later, HBV-DNA was negative, no hepatic disorder was observed, and chemotherapy for the primary disease could be carried out as planned. The key risk factors for HBVr are classified into three categories: (1) host factors, (2) virologic factors, and (3) type and degree of immunosuppression. Host factors include male, age, presence of cirrhosis, and type of cancer, (e.g., lymphoma). Virological factors were HBV-DNA at baseline and HBe antigen positivity. The type and degree of immunosuppression are associated with HBV coinfection with HCV or human immunodeficiency virus [11,17,39,40]. Another factor that exacerbates immunosuppression is the use of reagents and the frequency of chemotherapy. Chemotherapy, including rituximab for lymphoma, is a known risk factor for HBVr [37]. Furthermore, second- or third-line chemotherapy has been reported as a risk factor for HBVr [41]. The more frequent and longer the duration of chemotherapy, the more severe the immunosuppression in the host. Therefore, the duration of chemotherapy may be considered a factor for HBVr. In contrast, the treatment of HBV with NUCs in patients with HBVr was initiated prior to the appearance of a hepatic disorder due to HBV-DNA reactivation probably because the alert system made the attending physician understand that HBV-DNA monitoring is necessary not only within 3 months after the start of immunosuppressive drug therapy but also thereafter. Therefore, this alert system will be revised to a system that monitors HBV-DNA every 3 months in the future.

The percentage of patients in the HBV remission phase after the introduction of this alert system was significantly higher than that before because it improved the HBV-DNA measurement rates. The system recommended measuring HBV-DNA in HBsAg-negative patients with anti-HBs and/or anti-HBc. The HBV-DNA measurement rate in these patients improved significantly after the introduction of the system, and their titers were almost negative. In other words, these patients were mostly diagnosed in the HBV-remission phase. Thus, the alert system seemed to have increased the percentage of patients in the HBV remission phase by improving the HBV-DNA measurement rate in patients receiving immunosuppressive drug therapy.

The most important factor in preventing HBVr is the identification of HBV-DNA-positive patients. One reason for this is the presence of immune-escape mutants of HBV that do not produce HBsAg [42]. HBsAg immune-escape mutants are often caused by missense mutations involving only one amino acid residue rather than the insertion or deletion of multiple residues. HBsAg escape was first discovered in a follow-up study of pediatric vaccination in 1988, revealing that vaccinated children with a strong antibody response to HBsAg could still become HBsAg-positive [43]. The first mutation associated with this phenomenon was the replacement of the Gly residue at position 145 with the Arg residue (G145R) [44]. This mutation has since become the most widely reported vaccine avoidance variant, but reports of many other substitutions have been associated with HBsAg escape from vaccine-induced immunity. Only one patient had HBsAg immune escape mutation. She was an 83-year-old woman who had a negative HBsAg result on hematological examination prior to the introduction of treatment for marginal zone lymphoma. Anti-HBs and anti-HBc were measured, and both were positive; hence, we recommended HBV-DNA determination using the electronic alert. As HBV-DNA in this patient was positive at 2.7 log_10_ IU/mL, prophylactic treatment with entecavir was started immediately. Subsequently, chemotherapy was started. After 2 weeks, HBV-DNA was negative, and no hepatic disorder was observed during the course of treatment. The system guided HBV-DNA measurement in this patient, and the patient was immediately started on treatment with NUCs. Consequently, the patient was able to undergo chemotherapy without HBVr.

HBs-positive patients with positive HBV-DNA need to be treated with NUCs before initiating immunosuppressive drug therapy. In this study, four patients with extremely low HBsAg levels and negative for HBV-DNA were diagnosed to be in the remission phase. These patients did not receive NUCs and did not present HBVr within 3 months after the introduction of immunosuppressive drug therapy. Thus, among HBsAg-positive patients, some patients are negative for HBV-DNA and do not need to be treated with NUCs. Treatment with NUCs is a long-term treatment with some risk of adverse events, such as gastrointestinal symptoms. In this study, HBsAg levels were significantly higher in HBV-DNA-positive patients than in HBV-DNA-negative patients. Therefore, we examined the cut-off value of HBsAg to select the patients who needed treatment with NUCs among HBsAg-positive patients. The detection rate of HBV-DNA in HBsAg-positive patients was only 55.8% (25/43). The receiver operating characteristic (ROC) curve indicated that when the cutoff of HBsAg was set at 12.65 IU/mL (area under the ROC curve = 0.85), the detection rate of HBV-DNA-positive patients among HBsAg-positive patients improved to 72.4% (21/29) (Appendix A). Based on the understanding that the measurement of HBV-DNA is important, the changes in serum HBsAg cut-off levels in patients before immunosuppressive drug therapy may be considered in this alert system for HBVr.

Our study had two limitations. First, it was a single-center retrospective study. We plan to investigate the usefulness of this alert system in a multi-center study in the future. The second limitation is that the HBV-DNA levels of some patients under immunosuppressive drug therapy were not measured/monitored because these patients were transferred to other hospitals or died within 3 months. Therefore, the rate of HBV-DNA level measurement and HBV monitoring in patients in the remission phase was not 100% in this study. We plan to investigate the follow-up data regarding HBV-DNA levels in the patients who were transferred to other hospitals, in the future.

Most people who have received the HBV vaccine have anti-HBs. However, the vaccination against HBV has not been included in routine immunization in Japan. HBV vaccination for children has been mandatory since 2016. Therefore, in this study, vaccinated individuals were rarely included, and the presence of anti-HBs is unlikely to be the effect of HBV vaccination. The introduction of an alert system for HBV-DNA monitoring for every patient with HBsAg-negative/anti-HBs-positive or anti-HBc positive deserves cost-effectiveness evaluation in HBV low-prevalence regions.

## 5. Conclusions

This is the first report on an electronic alert system for HBVr that recommends HBV-DNA measurement. This alert system for HBVr improves HBV-DNA measurement rates in HBsAg-negative patients with anti-HBs and/or anti-HBc and in HBV-remission phase patients within 3 months after the immunosuppressive drug therapy. Moreover, patients with HBVr can be detected early and treated with NUCs before the onset of severe hepatitis. The benefit of introducing an alert system is to improve the HBV-DNA measurement rate for target patients. Moreover, the effect is expected to improve the HBV-DNA measurement rate not only within 3 months after the start of immunosuppressive drug therapy but also in terms of repeated monitoring every 3 months. This electronic alert system may also improve the rate of HBV-DNA measurement not only in target patients but also in other patients who are treated by the same physician. These results suggest that this alert system for HBVr is beneficial.

## Figures and Tables

**Figure 1 jcm-11-02446-f001:**
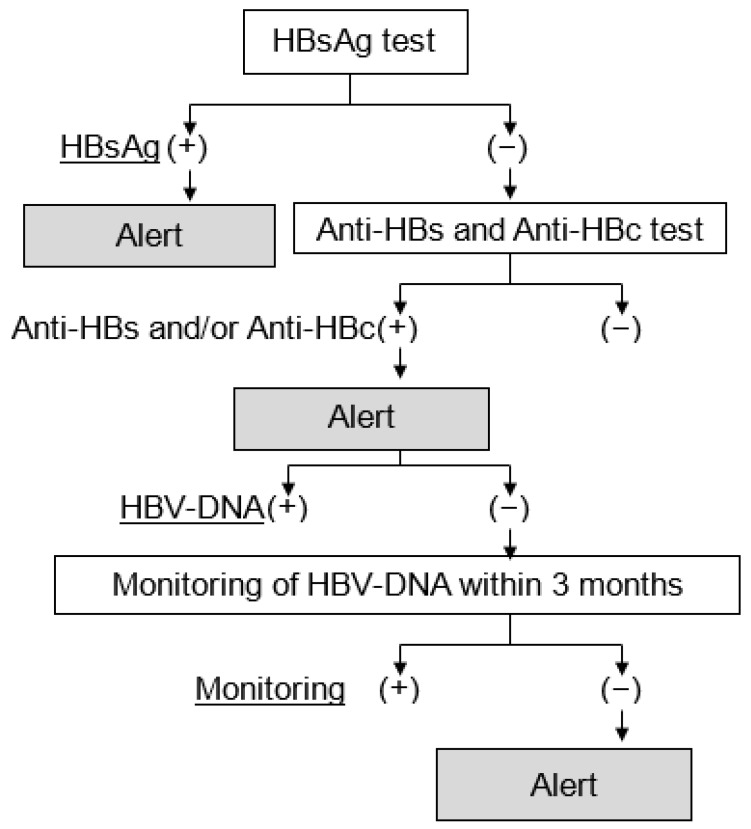
Schematic diagram of the electronic alert system to measure HBV-DNA in HBVr. Abbreviations: HBsAg, hepatitis B surface antigen; anti-HBs, antibody to hepatitis B surface antigen; anti-HBc, antibodies to hepatitis B core antigen; HBV-DNA, hepatitis B virus DNA; HBVr, HBV reactivation.

**Figure 2 jcm-11-02446-f002:**
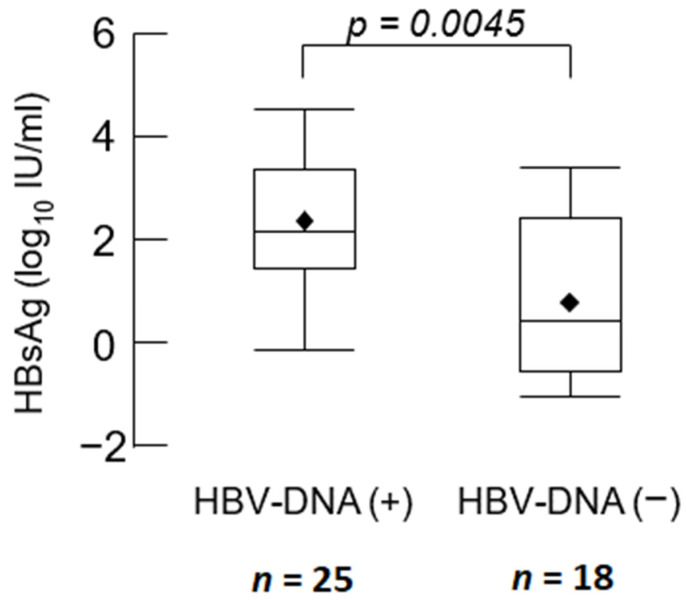
HBsAg levels in HBV-DNA positive and negative patients. Abbreviations: HBsAg, hepatitis B surface antigen; HBV-DNA, hepatitis B virus DNA.

**Table 1 jcm-11-02446-t001:** Baseline characteristics of patients before and after the introduction of the alert system.

	Before	After	*p*-Value
Duration	August 2015 to July 2018	October 2019 to April 2020	
Number of patients	4417	912	
Patients who received chemotherapy	2874 (65.1%)	588 (64.5%)	0.73
Patients who received other immunosuppressive therapy	1543 (34.9%)	324 (35.5%)	
Gender (male/female)	2039/2378	429/483	0.63
Age (median, 95% CI)	65 (59.5–60.6)	66 (59.9–62.2)	0.09
HBsAg-positive patients	42 (1.0%)	9 (1.0 %)	0.92
HBsAg-negative patients	4374 (99.0%)	903 (99.0%)	
Patients with anti-HBs and/or anti-HBc	917 (20.8%)	219 (24.0%)	0.03
Patients with anti-HBs alone	185	32	
Patients with anti-HBc alone	205	42	
Patients with anti-HBs and anti-HBc	527	145	
HBV-remission patients	704 (15.9%)	188 (20.6%)	<0.01

Abbreviations: CI, confidence interval; HBsAg, hepatitis B surface antigen; anti-HBs, anti-hepatitis B surface antibody; anti-HBc, anti-hepatitis B core antibody.

**Table 2 jcm-11-02446-t002:** Measurement rates of HBV-DNA in patients before and after the introduction of the alert system.

	Before	After	*p*-Value
**HBsAg-positive patients**	**(*n* = 42)**	**(*n* = 9)**	
Measurement for HBV-DNA	35	8	0.68
HBV-DNA-positive patients	21 (60.0%)	4 (50.0%)	0.76
**HBsAg-negative patients**	**(*n* = 4374)**	**(*n* = 903)**	
Patients with anti-HBs and/or anti-HBc	917	219	
Measurement for HBV-DNA before therapies	705 (76.9%)	188 (85.8%)	<0.01
HBV-DNA-positive patients	1 (0.1%)	0 (0%)	0.61
**HBV-remission patients with therapies**	**(*n* = 704)**	**(*n* = 188)**	
Patients without HBV-DNA	704	188	
Monitoring of HBVDNA within 3 months after therapies	305 (43.3%)	104 (55.3%)	<0.01
HBV-DNA-positive patients	2 (0.7%)	0 (0%)	0.46

Abbreviations: HBsAg, hepatitis B surface antigen; HBV-DNA, hepatitis B virus deoxyribonucleic acid; anti-HBs, anti-hepatitis B surface antibody; anti-HBc, anti-hepatitis B core antibody.

**Table 3 jcm-11-02446-t003:** Clinical characteristics of patients with HBV reactivation in the remission phase.

Age	Gender	Background Disease	Treatment for Background Disease	Duration of Treatment [days]	Treatment for HBV	Pre-Anti-HBs [mIU/mL]	Pre-Anti-HBc [S/CO]	Post-HBV-DNA [log_10_ IU/mL]	Post-ALT [IU/L]
76	Male	Renal cell carcinoma	GC	84	TAF	0.17	9.39	1.72	41
81	Male	Bladder cancer	Low dose GC	18	ETV	26.72	1.77	1.75	7

Abbreviations: HBV, hepatitis B virus; anti-HBs, anti-hepatitis B surface antibody; anti-HBc, hepatitis B core antibody; HBV-DNA, hepatitis B virus deoxyribonucleic acid; ALT, alanine aminotransferase; GC, gemcitabine and cisplatin; TAF, tenofovir alafenamide; ETV, entecavir.

**Table 4 jcm-11-02446-t004:** Measurement rates of HBV-DNA in patients undergoing chemotherapy before and after the introduction of the alert system.

	**Before**	**After**	***p*-Value**
Number of patients who were tested	2874	588	
Gender (male/female)	1477/1397	311/277	0.51
Age (median, 95% CI)	63 (63.2–64.1)	65 (64.1–66.0)	0.03
Measurement of HBV-DNA.			
	**Before**	**After**	***p*-Value**
**HBsAg-positive patients**	**(*n* = 32)**	**(*n* = 6)**	
Measurement for HBV-DNA	28	5	0.79
HBV-DNA-positive patients	16 (57.1%)	1 (20.0%)	0.12
**HBsAg-negative patients**	**(*n* = 2842)**	**(*n* = 582)**	
Patients with anti-HBs and/or anti-HBc	639	143	0.27
Measurement for HBV-DNA before chemotherapy	490 (76.7%)	127 (88.8%)	<0.01
HBV-DNA-positive patients	1 (0.2%)	0 (0%)	0.61
**HBV-remission patients undergoing chemotherapy**	**(*n* = 489)**	**(*n* = 127)**	
Patients without HBV-DNA	489	127	<0.01
Monitoring of HBVDNA within 3 months after chemotherapy	179 (36.6%)	65 (51.2%)	<0.01
HBV-DNA-positive patients	2 (1.1%)	0 (0%)	0.39

Abbreviations: CI, confidence interval; HBsAg, hepatitis B surface antigen; HBV-DNA, hepatitis B virus deoxyribonucleic acid; anti-HBs, anti-hepatitis B surface antibody; anti-HBc, anti-hepatitis B core antibody.

**Table 5 jcm-11-02446-t005:** Measurement rates of HBV-DNA in patients who received other immunosuppressive therapies before and after the introduction of the alert system.

	**Before**	**After**	** *p-* ** **Value**
Number of patients tested	1543	324	
Gender (male/female)	562/981	118/206	0.99
Age (median, 95% CI)	59 (52.0–54.4)	60 (51.6–56.7)	0.70
Measurement of HBV-DNA			
	**Before**	**After**	** *p-* ** **Value**
**HBsAg-positive patients**	**(*n* = 10)**	**(*n* = 3)**	
Measurement for HBV-DNA	7	3	0.28
HBV-DNA-positive patients	5 (71.4%)	3 (100%)	0.30
**HBsAg-negative patients**	**(*n* = 1532)**	**(*n* = 321)**	
Patients with anti-HBs and/or anti-HBc	278	76	0.02
Measurement for HBV-DNA	215 (77.3%)	61 (80.3%)	0.59
HBV-DNA-positive patients	0 (0%)	0 (0%)	N.A.
**HBV-remission patients with therapies**	**(*n* = 215)**	**(*n* = 61)**	
Patients without HBV-DNA	215	61	0.59
Monitoring of HBVDNA within 3 months after therapies	126 (58.6%)	39 (63.9%)	0.45
HBV-DNA-positive patients	0 (0%)	0 (0%)	N.A.

Abbreviations: CI, confidence interval; HBsAg, hepatitis B surface antigen; HBV-DNA, hepatitis B virus deoxyribonucleic acid; anti-HBs, anti-hepatitis B surface antibody; anti-HBc, anti-hepatitis B core antibody; NA, not available.

## Data Availability

Data are available on request owing to restrictions such as privacy and ethics. Data presented in this study are available upon request from the corresponding authors. Data were not made publicly available because of the presence of personal information.

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
