# Peer review of "Effect of an Electronic Alert System on Hepatitis B Virus Reactivation in Patients Receiving Immunosuppressive Drug Therapy"

_jcm, 2022, doi:10.3390/jcm11092446_

Round 1

Reviewer 1 Report

  1. Replace reference 5 in the first paragraph of the Introduction to a better article suitable for its statement on annual mortality of HBV infection.
  2. Please make sure that reference 23 is appropriate for citation in this manuscript since it was the JSH guidelines for HCV management instead of HBV.
  3. Please make sure that the number was 25 (positive for both HBsAg and HBV-DNA) instead of 24 in the final paragraph of page 5, because in the figure 2 HBV-DNA (+) n = 25.
  4. Correct figure 2 HBV-DNA (-) n = 18 (instead of 19) since HBV-DNA were measured in 43 HBsAg (+) subjects while 25 were positive.
  5. Why the number of HBsAg (+) 42 plus HBsAg (-) 4374 was not equal to 4417 in total enrolled patients before the alert system was introduced (Table 1)?
  6. In the first paragraph of page 11, ‘the detection rate of HBV-DNA in HBsAg-positive patients was only 55.8% (24/43)’? -> why not 25/43?? And after adjusting HBsAg detection cut-off to 12.65 IU/mL, the HBV-DNA-positive rate in HBsAg-positive subjects improved to 72.4% (21/29), please show this data in the tables. I am wondering that since this ‘new cut-off’ is more sensitive for HBV-DNA detection, why the authors did not re-calculate the case numbers in subjects prior to the introduction of the alert system.
  7. In table 2, no HBV-DNA (+) case was identified after the introduction of the alert system in 188 patients receiving HBV-DNA measurement before therapies, so please clarify when the 83-year-old lady with marginal zone lymphoma was detected to have 2.7 log HBV-DNA (mentioned in page 10)?
  8. Unify the spelling for ‘HBsAg’, ‘HBsAg-positive’, and ‘HBV-DNA” in the whole context.

The importance of HBV-DNA periodical monitoring (q3M) in HBsAg-negative subjects undergoing systemic chemotherapy or immunosuppressants was recommended only in HBVr high-risk populations in the most dominant international liver societies at present. The introduction of the alert system for HBV-DNA monitoring for every HBsAg-negative/anti-HBc or anti-HBs-positive patient deserves cost-effectiveness evaluation before it becomes a standard-of-care in HBV low-prevalence regions or countries. 

Author Response

Response to Reviewer 1 comments

1. Replace reference 5 in the first paragraph of the Introduction to a better article suitable for its statement on annual mortality of HBV infection.

We replaced the reference 5 about the mortality of HBV infection from World Health Organization.

2. Please make sure that reference 23 is appropriate for citation in this manuscript since it was the JSH guidelines for HCV management instead of HBV.

We replaced the reference 23 about the JSH guideline of HBV infection.

3. Please make sure that the number was 25 (positive for both HBsAg and HBV-DNA) instead of 24 in the final paragraph of page 5, because in the figure 2 HBV-DNA (+) n = 25.

In response to your comment, we make sure and changed the number “those who were positive for HBsAg and HBV-DNA (n = 25)” (page 6, line 204)

4. Correct figure 2 HBV-DNA (-) n = 18 (instead of 19) since HBV-DNA were measured in 43 HBsAg (+) subjects while 25 were positive.

In response to your comment, we make sure and changed the number “Patients who were positive for HBsAg and negative for HBV-DNA (n = 18)” (page 6, lines 205-6) and HBV-DNA (-) n = 18 in figure 2                                                        

5. Why the number of HBsAg (+) 42 plus HBsAg (-) 4374 was not equal to 4417 in total enrolled patients before the alert system was introduced (Table 1)?

Because one patient did not measure HBsAg, number was not equal to 4417.

6. In the first paragraph of page 11, ‘the detection rate of HBV-DNA in HBsAg-positive patients was only 55.8% (24/43)’? -> why not 25/43?? And after adjusting HBsAg detection cut-off to 12.65 IU/mL, the HBV-DNA-positive rate in HBsAg-positive subjects improved to 72.4% (21/29), please show this data in the tables. I am wondering that since this ‘new cut-off’ is more sensitive for HBV-DNA detection, why the authors did not re-calculate the case numbers in subjects prior to the introduction of the alert system.

This retrospective study used HBsAg test kit for medical use (ARCHITECT® HBsAg measurement kits, Abbott Japan Co., Ltd., Chiba, Japan). This kit defines the positive as 0.05 IU/ml or more in the manual and this threshold was used in this study. New cut-off value is intended to improve this alert system in the future, and we will conduct the new research using the new cut-off value. In response your comment, we changed the number from 24/43 to 25/43 (Page 11, line 412) and inserted the supplemental table 1.

7. In table 2, no HBV-DNA (+) case was identified after the introduction of the alert system in 188 patients receiving HBV-DNA measurement before therapies, so please clarify when the 83-year-old lady with marginal zone lymphoma was detected to have 2.7 log HBV-DNA (mentioned in page 10)?

The 83-year-old lady with marginal zone lymphoma belonged to the group before the introduction of the system. And, she was only one patient with HBsAg-negative, anti-HBs-positive, anti-HBc-positive, and HBV-DNA-positive before therapies.

8. Unify the spelling for ‘HBsAg’, ‘HBsAg-positive’, and ‘HBV-DNA” in the whole context.

In response to your comments, we unified these words.

The importance of HBV-DNA periodical monitoring (q3M) in HBsAg-negative subjects undergoing systemic chemotherapy or immunosuppressants was recommended only in HBVr high-risk populations in the most dominant international liver societies at present. The introduction of the alert system for HBV-DNA monitoring for every HBsAg-negative/anti-HBc or anti-HBs-positive patient deserves cost-effectiveness evaluation before it becomes a standard-of-care in HBV low-prevalence regions or countries. 

We inserted the sentences about cost-effectiveness evaluation in page 11, lines 431-3.

“The introduction of alert system for HBV-DNA monitoring for every patient with HBsAg-negative/ anti-HBs-positive or anti-HBc positive deserves cost-effectiveness evaluation in HBV low-prevalence regions.”

Reviewer 2 Report

jcm-1652752-peer-review-v1, review on

Can you modify this sentence, as I think it’s not the patients’ fault but providers’:

“Although guidelines for HBVr 14 have been proposed by several academic societies, some patients do not follow them and consequently develop HBVr, which results in death.”

The authors claim lack of report on alert systems, they may want to put their paper into perspective with PMID: 32080253, PMID: 31314189, PMID: 30202826, PMID: 26819740

This may need some rewording, as it’s not 5% of those who seroconvert who become chronic line 40-43, and it’s not the infection that recovers but the individual:

“Acute HBV infection is usually subclinical in adults. In immunocompetent adults, acute HBV infection spontaneously recovers with hepatitis B surface antigen (HBsAg) to antibody to hepatitis B surface antigen (anti-HBs) seroconversion, and only 5% of these adults develop chronic HBV infection.“

Do the authors believe HBV persists only up to 10 years in the liver? See following sentence:

HBV persists in the liver up to 10 years after acute HBV infection [12].

Why do you feel you need to monitor HBsAb positive individuals, who most likely were vaccinated?

As you want to test after 90 days of start of therapy for monitoring, a better window may be day 60-120 vd. Less than 91 days. What about repeated testing, as reactivation usually occurs after months or even years of treatment.

Give info on HBc alone, HBs alone and HBs and HBc positive.

What was the HBV status of the 2 cases? Were these the only 2 reactivations among all patients?

The confidence intervals for Age would be more appropriate than p-value in the sentence:

The mean age was 63.7 ± 13.4 years before and 64.9 ± 13.2 years after the introduction of the system, indicating a significantly higher age after the introduction of the system (p= 0.0253) (line 221 to 223.)

The tables need to be more easily understandable, please reorganize them, and limit to relevant information.

You may benefit from English language editor, and

Author Response

Response to Reviewer 2 comments

Can you modify this sentence, as I think it’s not the patients’ fault but providers’:

“Although guidelines for HBVr 14 have been proposed by several academic societies, some patients do not follow them and consequently develop HBVr, which results in death.”

In response to your comment, we changed from patients to providers (page 1, line 15)

The authors claim lack of report on alert systems, they may want to put their paper into perspective with PMID: 32080253, PMID: 31314189, PMID: 30202826, PMID: 26819740

 In response to our comment, we added the references and changed the sentence to “Some reports on electronic alerts for the screening of Hepatitis C virus (HCV) and HBV infection are available [28,29]; however, electronic alerts for HBVr are insufficient [30,31].” (Page 2, lines 72-74).

This may need some rewording, as it’s not 5% of those who seroconvert who become chronic line 40-43, and it’s not the infection that recovers but the individual:

“Acute HBV infection is usually subclinical in adults. In immunocompetent adults, acute HBV infection spontaneously recovers with hepatitis B surface antigen (HBsAg) to antibody to hepatitis B surface antigen (anti-HBs) seroconversion, and only 5% of these adults develop chronic HBV infection.“

 In response to your comment, we changed from “recover” to “change” in page 1, line 41.

Do the authors believe HBV persists only up to 10 years in the liver? See following sentence:

HBV persists in the liver up to 10 years after acute HBV infection [12].

In response to your comment, we added the word “more than 10 years after acute infection” in page 2, line 54.

Why do you feel you need to monitor HBsAb positive individuals, who most likely were vaccinated?

As your pointed out, most people who have been received HBV vaccine have anti-HBs. However, in Japan, HBV vaccine has long not been included in routine immunization. Since 2016, HBV vaccination for children has become mandatory. Therefore, the patients in this study rarely include those patients, and anti-HBs is unlikely to be the effect of HBV vaccination.

We inserted these sentences in discussion (page 11, lines 427 - 430).

As you want to test after 90 days of start of therapy for monitoring, a better window may be day 60-120 vd. Less than 91 days. What about repeated testing, as reactivation usually occurs after months or even years of treatment.

As same as your pointed out, we will revise this system that monitor HBV-DNA every three months in the future. (Page 10, line 369-370)

Give info on HBc alone, HBs alone and HBs and HBc positive.

In response to your comment, we added the data of patients with anti-HBs alone, anti-HBc alone, and both in the table 1.

What was the HBV status of the 2 cases? Were these the only 2 reactivations among all patients?

In response to your comment, among patients with remission phase, there are only two patients who have reactivated of HBV within three months after immunosuppressive drug therapy in this study.

The confidence intervals for Age would be more appropriate than p-value in the sentence:

The mean age was 63.7 ± 13.4 years before and 64.9 ± 13.2 years after the introduction of the system, indicating a significantly higher age after the introduction of the system (p= 0.0253) (line 221 to 223.)

 In response to your comment, we changed from tables and sentences as below (page 7, lines 256-257).

“The median age with 95% confidence interval was 63 (63.2-64.1) before and 65 (64.1-66.0) after the introduction of the system.”

The tables need to be more easily understandable, please reorganize them, and limit to relevant information.

 In response to your comments, we reorganize tables and limit to some data.

You may benefit from English language editor, and

In response to your comment, we requested the editing of this manuscript for English language. This manuscript has been edited by Editage (www.editage.com).

Round 2

Reviewer 1 Report

The authors revised the manuscript point-to-point according to my previous comments, I have no further comments.